# Effects of a Combined Community Exercise Program in Obstructive Sleep Apnea Syndrome: A Randomized Clinical Trial

**DOI:** 10.3390/jcm8030361

**Published:** 2019-03-14

**Authors:** Rodrigo Torres-Castro, Jordi Vilaró, Joan-Daniel Martí, Onintza Garmendia, Elena Gimeno-Santos, Bárbara Romano-Andrioni, Cristina Embid, Josep M. Montserrat

**Affiliations:** 1Servicio de Neumología, Unidad del Sueño, Hospital Clínic de Barcelona, 08036 Barcelona, Spain; onintzag@gmail.com (O.G.); cembid@clinic.cat (C.E.); jmmontserrat2@gmail.com (J.M.M.); 2Departamento de Kinesiología, Facultad de Medicina, Universidad de Chile, 8380453 Santiago, Chile; 3Facultad de Ciencias de la Salud Blanquerna. Global Research on Wellbeing (GRoW)), Universitat Ramon Llull, 08025 Barcelona, Spain; jordivc@blanquerna.url.edu; 4Unidad de Cuidados Intensivos en Cirugía Cardiovascular, Instituto Clínico Cardiovascular, Hospital Clínic de Barcelona, 08036 Barcelona, Spain; jdmartibcn@gmail.com; 5Respiratory Institute, Hospital Clínic de Barcelona, 08036 Barcelona, Spain; elenagimeno@gmail.com; 6Institut d’Investigacions Biomèdiques August Pi i Sunyer (IDIBAPS), 08036 Barcelona, Spain; 7Servicio de Endocrinología y Nutrición, Hospital Clínic de Barcelona, 08036 Barcelona, Spain; bromano@clinic.cat

**Keywords:** physical training, oropharyngeal exercises, apnea-hypopnea syndrome, urban-walking program

## Abstract

Physical activity is associated with a decreased prevalence of obstructive sleep apnea and improved sleep efficiency. Studies on the effects of a comprehensive exercise program in a community setting remain limited. Our objective was to investigate the effects of a combined physical and oropharyngeal exercise program on the apnea-hypopnea index in patients with moderate to severe obstructive sleep apnea. This was a randomized clinical trial where the intervention group followed an eight-week urban-walking program, oropharyngeal exercises, and diet and sleep recommendations. The control group followed diet and sleep recommendations. A total of 33 patients were enrolled and randomized and, finally, 27 patients were included in the study (IG, 14; CG, 13) Obstructive sleep apnea patients were analyzed with a median age of 67 (52–74) and median apnea-hypopnea index of 32 events/h (25–41). The apnea-hypopnea index did not differ between groups pre- and post-intervention. However, in intervention patients younger than 60 (*n* = 6) a reduction of the apnea-hypopnea index from 29.5 (21.8–48.3) to 15.5 (11–34) events/h (*p* = 0.028) was observed. While a comprehensive multimodal program does not modify the apnea-hypopnea index, it could reduce body weight and increase the walking distance of patients with moderate to severe obstructive sleep apnea. Patients younger than 60 may also present a decreased apnea-hypopnea index after intervention.

## 1. Introduction

In obstructive sleep apnea (OSA) intermittent collapse of the upper airway occurs during sleep, which leads to transient asphyxia [1]. This disease affects between 9 and 38% of general adult population, constituting a public health concern [2]. OSA is associated with a wide range of health consequences, including daytime sleepiness, cognitive impairment, and metabolic and cardiovascular diseases [3]. The recommended first-line of treatment for OSA is continuous positive airway pressure (CPAP) [3,4,5], a therapy that prevents apnea events by maintaining the airways open. Treatment with CPAP is cost-effective [6], decreases morbidity and mortality due to cardiovascular diseases [7], and reduces the risk of drowsiness-related traffic accidents [8].

In the last decade, comprehensive treatments that include physical [9,10,11] and oropharyngeal [12,13,14] exercises have been suggested for lowering the apnea-hypopnea index (AHI) of OSA patient population. Indeed, physical training programmes are associated with decreased OSA prevalence, lower AHI scores, improved sleep efficiency, and less daytime sleepiness [15,16]. While the mechanisms underlying these beneficial effects in OSA patients are not clearly understood, it is known that exercise can reduce body weight and fatty tissue contents, which have been related to significant reductions in the AHI [17]. Importantly, approximately 60–70% of OSA patients are overweight or obese [18], and these conditions constitute independent risk factors for the occurrence of OSA. However, several studies have found that the beneficial effects of exercise on OSA are not related to a reduced body weight [15,16,19,20]. Instead, some research suggests that apnea occurrence could potentially be reduced by decreasing lower extremity edema through physical exercise [21].

As mentioned, recent scientific evidence [13,14,22] supports the use of oropharyngeal exercises commonly used in speech and dysphagia rehabilitation for treating OSA. Possible benefits of these exercises include a reduction in neck circumference, snoring, subjective sleepiness, and AHI, as well as an improved quality of life. Specifically, the genioglossus and pharyngeal musculatures, which play an important role in OSA, can be trained through oropharyngeal exercises [13].

Despite the potential benefits of physical exercise, OSA patients rarely adhere to rehabilitation programs, which commonly require scheduled hospital visits or the use of sophisticated training equipment (e.g., treadmill, cycle ergometer) [10,20,23,24] not always available to the patient outside of the rehabilitation program

The aim of this study was to investigate the effects of a combined physical and oropharyngeal exercise program on AHI scores in patients with moderate to severe OSA that were recommended CPAP therapy.

## 2. Experimental Section

### 2.1. Participants

Study participants were recruited from the Sleep Unit at the Hospital Clinic of Barcelona, Spain. The institutional ethics committee approved the study, and informed written consent was obtained from all participants prior to the start of the study.

Patients in the Sleep Unit were screened daily to identify adults between 40 and 85 years-old that were recently diagnosed with moderate or severe OSA (AHI ≥ 15 events/h), were CPAP candidates according to medical criteria, and were living in Barcelona. Exclusion criteria included a body mass index (BMI) > 40 kg/m^2^, somnolence affecting daily physical activities, smokers, neuromuscular disorders affecting buccal musculature, any chronic disease associated with medically-prescribed avoidance of physical exertion, or any muscular-skeletal alteration that would impair executing physical exercise. None of the subjects were CPAP users during this study.

### 2.2. Study Design

This study was a randomized controlled trial (NCT02482480), and Consolidated Standards of Reporting Trials (CONSORT) recommendations for non-pharmacological studies were followed. Participating patients were randomized after baseline assessment by an external investigator into one of two arms (1:1 ratio) using statistical software in varying blocks of two, four, or six. The lead physician and technician analyzing the sleep studies were unaware of patient assignments during the study.

### 2.3. Intervention

Participants in the control group received general recommendations regarding diet, sleep hygiene, and physical activity, such as walking for 30 min at least three times a week while maintaining the fastest possible pace.

In contrast, participants in the intervention group followed a comprehensive community program that included general physical activity, oropharyngeal exercises, and diet and sleep hygiene recommendations. Physical activity consisted in 30 min of walking along publicly accessible urban park tracks designed and standardized for chronic obstructive pulmonary disease patients [25]. Participants were encouraged to maintain a high walking intensity by achieving 60–80% of the theoretical maximal heart rate, as controlled by a heart rate monitor (Polar RS800, Kempele, Finland). The intervention program lasted eight weeks, with three walking sessions/week. A physiotherapist supervised all sessions during the first two weeks and one session/week from the third until the last week of the program (i.e., 12 supervised sessions, 24 sessions total). In unsupervised sessions, the physiotherapist called each patient to ask if there was any inconvenience in carrying out the protocol of general exercise. If the patient could not train, a new session was rescheduled.

Additionally, during the first supervised walking sessions, the physiotherapist provided educational information about factors affecting OSA severity (Appendix A), as well as recommendations about sleep hygiene, diet, and weight control. Moreover, patients received a document with general advice about diet control prepared by a nutritionist.

Oropharyngeal exercises were based on the studies by Guimarães et al. [14] and Burkhead et al. [26] (Appendix A). The selected oropharyngeal exercises are used to treat speech-language pathologies and included soft palate, tongue, and facial muscle exercises as well as stomatognathic function exercises. Patients were instructed to practice four exercises every day at least five days per week. A video of each exercise was provided to ensure proper task realization. Weekly, the physiotherapy in charge of the program ask the patients about the exercises accomplishment, difficulties, doubts, etc. High session attendance was defined as attending at least 80% of the sessions. The patients completed a compliance chart every day. If one day they could not perform the exercises, they would recover with a weekend day.

The dietary recommendations were based on the promotion of a low-carbohydrate diet, compliance with meal times and the elimination of foods that affect sleep conciliation (e.g., coffee, cola drinks, alcohol) (Appendix A). Sleep hygiene recommendations include, among others, avoiding the use of screens before sleep, avoid carrying out intense physical exercise before sleep, avoid hearty meals. We explained all this recommendation in one individualized session of 25 min with all patients independent of the group assigned.

### 2.4. Measurements

Before and after intervention, participants were required to complete a polysomnography, respiratory polygraphy, and six-minute walking test (6MWT), undergo anthropometrical measurements, and respond to symptoms and quality of life questionnaires.

Standard laboratory polysomnography (Somté PSG, Compumedics Limited 2006, Abbotsford, Victoria, Australia) was performed according to the technical specifications of the American Academy of Sleep Medicine [27]. The recorded variables were an electroencephalogram (C3–A2, C4–A1, O1–A2, O2–A1), electrooculogram (two channels), chin and leg electromyograms, and electrocardiogram. Frontal electrodes were not used. Respiratory variables were measured by linearized nasal pressure prongs and the flow waveform of oronasal thermal signals. Respiratory effort signals were measured through inductive bands that recorded ribcage and abdominal movements. Oxygen saturation, body position, and snoring were also registered. The polysomnographic results of each participant were manually analyzed by a blinded technician.

Respiratory polygraphy was performed according to the technical specifications of the American Academy of Sleep Medicine [27] with an Embletta portable diagnostic system (ResMed. Sydney, Australia). Measurements included a snoring sensor, nasal thermistor, and nasal pressure cannula to register airflows; thoracic and abdominal belts to assess ribcage and abdominal movements; an electrocardiogram; an actigraph to detect positions; oxygen saturation; and heart pulse.

All patients were evaluated through the same test (polysomnography or respiratory polygraphy) before and after the intervention. The pulmonologist’s medical criteria determinated which test was chosen for each patient. For both the polysomnography and respiratory polygraphy, apnea or hypopnea were analyzed and scored according to the following criteria: hypopneas were defined as a ≥30% decrease in airflow signal amplitude lasting ≥10 s and accompanied by ≥3% oxygen desaturation, while apneas were defined as a ≥90% decrease in airflow signal amplitude lasting ≥10 s. The oxygen desaturation index was used to establish the quantity of oxygen desaturations ≥3%. Furthermore, both tests defined moderate OSA as an AHI with 15–29.9 events per hour of sleep and severe OSA as ≥30 events per hour of sleep.

Anthropometric measurements included height and weight, with which the BMI calculated. Neck, waist, and hip circumferences were obtained using a cloth measuring tape graduated in centimeters.

The 6MWT was carried out following regulations established by the Sociedad Española de Neumología y Cirugía Torácica [28]. The test was repeated twice, and participants were provided with standardized information and encouragement. The better distance of the two tests was used for analyses. The obtained values were compared to reference values using the European population reference equations described by Troosters et al. [29].

Participants completed the Spanish version of validated questionnaires for people with OSA. Subjective sleepiness was assessed using the Epworth Sleepiness Scale [30], which contains eight items that evaluate the likelihood of falling asleep during typical daytime activities. This probability ranges from 0 (never) to 30 (high probability). Normal values range from 2 to 10, with scores >10 indicating daytime sleepiness. Participants also completed the Quebec Sleep Questionnaire [31], a specific questionnaire to assess health-related quality of life in OSA patients. The Quebec Sleep Questionnaire evaluates the impact of OSA through 32 items divided into five different domains (i.e., hypersomnolence, daytime symptoms, nighttime symptoms, emotions, and social interactions). Finally, participants were asked to complete the Hospital Anxiety and Depression (HAD) Scale [32]. This self-reported scale is used to assess anxiety and depression in patients with medical and surgical conditions and consists in seven items related to an anxiety subscale and seven items related to a depression subscale. Each item is scored on a four-point scale, ranging from 0 (as much as I always do) to 3 (not at all); resulting in maximum subscale scores of 21 for each depression and anxiety.

### 2.5. Statistical Analysis

A sample size of 11 patients in each group was needed to achieve 8-point differences in the AHI, with a statistical power of 80%, an α level of 0.05, and an estimated loss rate of 30%. The assumed difference was based on the study by Kline et al. [9], which evaluated a protocol of physical activity with AHI as the main variable. Additionally, we calculated the sample size for Guimaraes who conducted a protocol of oropharyngeal exercises with similar characteristics. In this case a sample size of 10 patients in each group was needed to achieve nine-point difference in the AHI [14].

Statistical analyses were performed using SPSS Statistics v.22 for Windows (IBM Corporation, Armonk, NY, USA). Data distribution was assessed through a Shapiro–Wilk test. Continuous variables are presented as the mean ± standard deviation or as the median and 25–75th percentiles, as appropriate. Differences between groups were evaluated using an unpaired Student’s *t*-test for normally distributed variables, a Mann–Whitney *U* test for non-parametric variables, or a Fisher’s exact test for comparing proportions. Paired observations were analyzed using either a paired Student’s *t*-test or Wilcoxon test. A *p* < 0.05 was considered statistically significant.

## 3. Results

A total of 33 patients were enrolled and randomized (October 2014 to August 2015). Two participants were excluded due to medical complications of a chronic disease during the course of the program, and four participants voluntarily abandoned the study. Finally, 27 patients (15 men) were included in the study (Figure 1).

The characteristics of participants were median age of 67 years old (52–74), weigth of 82 Kg (70–90), BMI of 29.4 Kg/m^2^ (26.3–34.6), and AHI of 32 events/h (25–41). The demographic and sleep characteristics of the population, according to assigned group, are presented in Table 1. The baseline characteristics of all measured variables were not significantly different between groups.

The AHI did not differ either within or between the intervention or control groups (Table 2). When the intervention gorup was divided by age using post hoc analysis, participants younger than 60-years-old (*n* = 6) obtained a significant post-intervention decrease in the AHI, going from 29.5 (21.8–48.3) to 15.5 (11–34) events/h (*p* = 0.028). In contrast, the AHI in subjects older than 60-years-old (*n* = 8) increased from 31.5 (21–38.8) to 43.5 (24.8–48) events/h (*p* = 0.017) (Figure 2).

After the intervention period, no significant changes were observed in the control group (Table 2). Conversely, intervention group participants significantly improved pre- vs. post-intervention, respectively, in regards to body weight (86 (80.5–90.5) vs. 85.3 (77–88.3) Kg; *p* = 0.003), BMI (31.3 (27.5–35) vs. 30.2 (27.3–34.7) Kg/m^2^; *p* = 0.003), waist circumference (108.5 (102.5–114) vs. 107 (99–113.5) cm; *p* = 0.022), and in the six-minute walking distance (6MWD) (548.2 ± 83.9 vs. 567.1 ± 85.3 m; *p* = 0.020). No significant changes were observed regarding other variables.

In the control group, four patients did not have comorbidities, nine patients had one comorbidity. In the intervention group. In the intervention group, five patients did not have comorbidities, seven patients had one comorbidity, and two patients had two comorbidities. If we divided by age range, in the group older than 60 years, two patients did not have comorbidities, four patients had one comorbidity, and two patients had two comorbidities. In the group over than 60 years, three patients did not have comorbidities, and three patients had one comorbidity.

The comorbidities founded were: depression, dyslipidemia, arterial hypertension, diabetes, asthma, urinary infection, ocular infection, previous non-Hodgkin lymphoma. One patient had bariatric surgery (control group), and two patients (one control group, and one intervention group older than 60 years) had a previous oncological disease. All comorbidities were under control and medical supervision.

## 4. Discussion

This study found that a comprehensive community program combining physical and oropharyngeal exercises does not reduce AHI scores in patients with moderate to severe OSA syndrome as compared to a control group. However, patients under 60-years-old did obtain a significant reduction in the AHI after program completion.

The present results are not in line with most previously published evidence. Indeed, most studies in OSA patients report a significant decrease in the AHI after completing a rehabilitation program based on aerobic exercise [9,10,20,22]. However, in these studies, all of the rehabilitation programs were completed during out-patient hospital appointments that were carried out under strict supervision and using specific training equipment. Moreover, all participants had a mean age <60 years old and only a moderate OSA. Similarly, results obtained in studies implementing oropharyngeal exercises also observed significant improvements in terms of the AHI [13,14,22], but, once again, patients were younger and had a more moderate AHI than participants in the present trial.

Neumannova and colleagues, founded that a pulmonary rehabilitation and oropharyngeal exercises improve anthropometric parameters of obesity and decrease the BMI in patients with OSA. The difference with our patients is the use of positive pressure [33]. We trained the patients without positive pressure (without indication or during the process to installation).

In this study, the effects of a multimodal program significantly differed between patients older and younger than 60 years old (Table 3). Severe OSA patients under 60 significantly reduced their AHI by completion of the comprehensive program, while patients older than 60 did not achieve a significant reduction in this variable. These results are consistent with those obtained by Desplan et al. [10] and Kline et al. [9], researchers who also observed a decrease in AHI when young severe OSA patients were trained.

Interestingly, patients over 60 years old in the present study evidenced worsened sleep-related characteristics, for example in the AHI. Nevertheless, participants did generally improve in parameters directly influenced by the interventions, such as in walking distance, body weight, and BMI. This phenomenon could be partially explained by the anatomical and neurogenic changes that occur due to aging and disease progression. Indeed, Edwards and colleagues suggested differing phenotypes between OSA patients younger and older than 60 [34]. Furthermore, the anatomical collapsibility of upper airways is important in older adults, mainly since the major upper airway muscle dilator can perform poorly (i.e., genioglossus) [34].

Saboisky and colleagues [35] conducted a study comparing motor unit potential in subjects older and younger than 55, finding that this potential was 32% longer in duration in older subjects. Furthermore, cross-sectional morphological studies on human muscle samples corroborate a decrease in fiber number with aging [36]. These studies support that motor unit number remains fairly constant up to 60, but a subsequent decline of 50% occurs between 60- and 80-years-old [36].

However, this progressive and generalized loss of skeletal muscle mass and strength associated with aging, sarcopenia, can definitely be trained in the elderly, In our results, the group older than 60 of the intervention group improved significantly 23 m the distance walked in the 6MWT (*p* = 0.036) demonstrated an effect of this protocol in the physical capacity. This is concordant with the literature that has shown the improvement in this test and other functional tests with different physical training protocols [37,38]. Although the majority of studies focused on improving physical capacity in sarcopenia include muscle strengthening and aerobic exercise and last at least 12 weeks [38]. Since there are several mechanisms to reduce apneas, the effect of general physical training and specific training of the oropharyngeal muscles its remains to elucidate and must be specifically studied to determine the improvement of each training in people over 60. Until now, most of the studies, with both types of exercises, have been done in a population younger than 60 years [9,13,14,20,22,23,24].

Although the AHI of the group over 60 years (31.5) is similar to the group under 60 years (29.5), it is striking that the group under 60 years has a CT90 of 4.5%; instead, the group over 60 years has a CT90 three times higher. This means that, from the functional point of view, the apneas of both groups are not equivalent and have different consequences in arterial oxygenation. In the case of the group over 60 the weight was 84 and the median reduction was 1.4 Kg (*p* = 0.027), and in the group under 60 years, the weight was 90 Kg and the reduction was 3.5 Kg (*p* = 0.042). The literature describes the diminution of IAH independent of the reduction of weight [15,16]. This point, not yet explored, in the protocols of exercise in patients with sleep apnea can certainly provide information to understand the mechanism that explains why two groups with significant weight loss have different results in terms of AHI.

Another important aspect to consider is the applied setting and monitoring program. Most prior studies required subjects to participate in highly supervised exercise programs that took place at hospitals or rehabilitation centers [9,20,22]. Conversely, the presently applied protocol supervised a maximum 50% of exercise sessions (i.e., one day per week supervision). Another relevant aspect of this clinical trial is that training was performed outside the hospital environment, using existing resources within urban park tracks. These urban park tracks were previously validated for chronic respiratory disease to set different exercise intensities [25]. The use of urban parks tracks could promote program adherence by ensuring that every patient has the chance to go to a park near his/her home, thereby preventing an innecessary avoidance of time and cost consumption while promoting physical activity. From a clinical point of view, this could help to reduce costs and involve patients to a greater degree in their own self-care, especially when health-system resources are limited. Our patients reported 100% of accomplishment in the prescribed exercise sessions, oropharyngeal and walking training. Finally, this training methodology should be interesting when considering that it can induce long-term effects and could help to change patient perspectives towards a healthier lifestyle.

Our reported results suggest that to improve program objectives and effects, patient age may be an important determinant to keep in mind when considering inclusion in a pulmonary rehabilitation program. The sleep apnea episodes of younger patients were clearly impactful; whereas older individuals still improved in regards to personal exercise capacity and body weight despite not improving according to the AHI. Further studies with a wide range of ages and patients should be conducted to confirm these results.

This study has some limitations. The short-term duration of the integrated exercise program limits the external validity of the obtained findings. A longer intervention period and long-term monitoring would be needed to evaluate if intervention program benefits persist over time in the absence of continued physical activity and oropharyngeal exercises. Therefore, there were no assurances that the patients completed the exercise session on the days in which the physiotherapist was absent. This could possibly explain the low impact of the combined intervention program. Nevertheless, an extensive monitoring strategy was not used so as to provide as-close-to-a-clinical-setting as possible. Indeed, if these types of interventions are to be applied in the general OSA population, realistic scenarios need to be tested to determine efficacy. On the other hand, the sample size was small. We design a protocol with sample size calculation to explore the results with a comprehensive program in the general population without age categories, and we do not expect found differences between patients upper or below 60 years. We decided to report this fact, because “open a new unexplored door”: the effects of exercise in patients older than 60 years. Until now, the literature has reported the improvement in the AHI in oropharyngeal exercises and general exercise. However, we need to redesign new protocols with individual types of exercise in older patients. Another limitation is that the same test was not used to evaluate all patients. However, PSG or RP was used according to the pulmonologist’s medical criteria. To minimize this limitation, the same tests were performed before and after the intervention, and all the evaluations were analyzed with the variables that the polygraph delivered. The sleep studies have a variability night-to-night and, in our design, we only take one test per patient. However, this happened in both groups.

## 5. Conclusions

A short-term, comprehensive and community-based program that combined physical and oropharyngeal exercises did not produce significant changes in the AHI in patients with moderate to severe OSA that were recommended CPAP therapy. However, the body weight decreased and walking distance capacity increased after the program. Notably, patients under 60 years old that followed the comprehensive intervention program significant decreased the AHI score.

Further larger studies are required to support these findings and, thus, establish the effects of community-based, comprehensive physical activity programs and oropharyngeal exercise guidelines in treating patients with moderate to severe OSA.

## Figures and Tables

**Figure 1 jcm-08-00361-f001:**
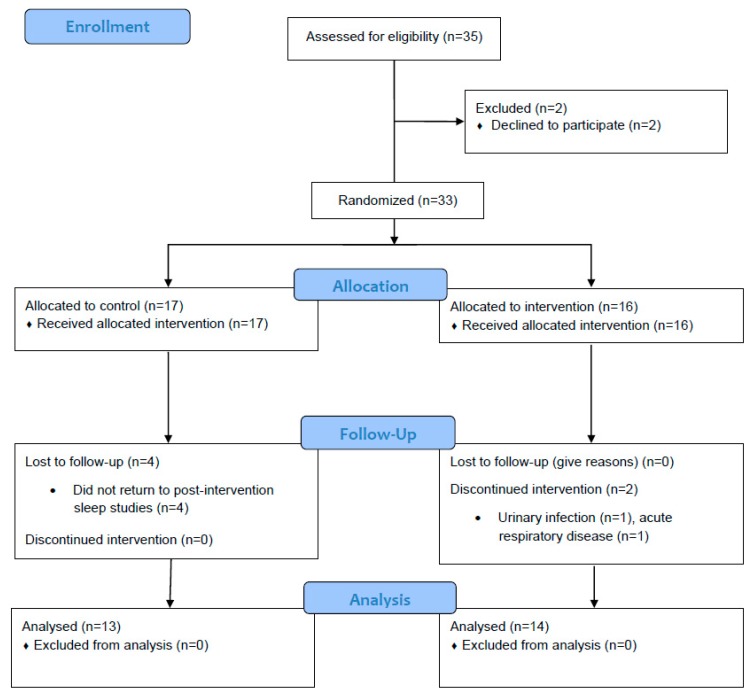
CONSORT flow chart of participants of study.

**Figure 2 jcm-08-00361-f002:**
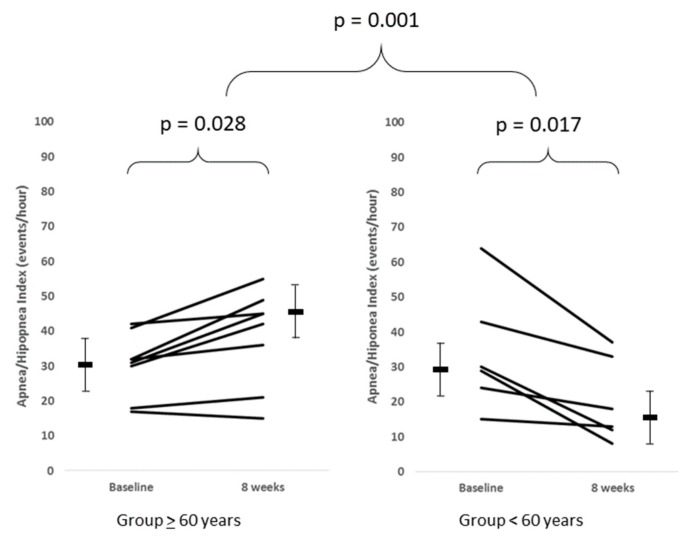
The apnea-hypopnea index before (i.e., baseline) and after a comprehensive eight-week protocol in patients with obstructive sleep apnea. Intragroup comparison before and after using a Wilcoxon test for skewness distribution. Intergroup comparisons for differences between variables were assessed using the Mann-Whitney test for skewness distribution.

**Table 1 jcm-08-00361-t001:** Baseline characteristics of study participants.

	Control (*n* = 13)	Intervention (*n* = 14)	*p*-Value
Anthropometric data			
Age, years	67 (53–74.5)	64.5 (51.8–74)	0.793
Males, %	57.1	53.8	
Body weight, Kg	82 (70–87.5)	86 (80.5–90.5)	0.116
Height, cm	166.6 ± 9.3	163.8 ± 8.3	0.458
BMI, Kg/m^2^	27.1 (25.1–35.9)	31.3 (27.5–35.0)	0.375
Neck circumference, cm	38 (37–41.5)	40 (37–41.5)	0.616
Waist circumference, cm	107 (97–114.5)	108.5 (102.5–114)	0.519
Hip circumference, cm	96 (91.5–107.5)	97 (93.5–109)	0.720
Polysomnographic data			
AHI/h	37 (25.5–43.5)	30.5 (22.5–41.3)	0.325
CT90	22 (3.5–28.5)	12 (4–21.3)	0.239
ODI/h	34 (29.5–44.5)	30 (22.3–39.8)	0.280
Exercise Capacity			
6MWD, m	529.8 ± 102.7	548.2 ± 83.9	0.550
Questionnaires			
ESS	8 (6.5–11)	8 (3–13)	0.756
QSQ			
Total score	172 (133–186.5)	180.5 (151.3–200)	0.430
Daytime Sleepiness	36 (29.5–37.5)	38 (29.3–40.3)	0.488
Diurnal symptoms	50 (43.5–62)	59.5 (45–66)	0.430
Nocturnal symptoms	35 (26.5–38)	35 (29.8–42.3)	0.430
Emotions	30 (19.5–31)	27 (21.8–29.3)	0.375
Social interactions	23 (17–24)	23 (21–25)	0.583
HAD			
Total score	14 (6–23.5)	9 (3.8–18.8)	0.550
Anxiety	8 (6–12.5)	6 (2.8–10)	0.325
Depression	6 (1–9)	2.5 (1–8)	0.720

Abbreviations: BMI: body mass index, AHI: Apnea Hypopnea Index, CT90: cumulative percentages of time spent at saturations below 90%, ODI: oxygen desaturation index, 6MWD: Six-minute walking distance, ESS: Epworth Sleepiness Scale; QSQ: Quebec Sleep Questionnaire, and HAD: Hospital Anxiety and Depression Score. Values are expressed as the mean ± SD if data is normally distributed or as the median (P25–P75) if data distribution is skewed.

**Table 2 jcm-08-00361-t002:** Changes between variables according to intervention groups.

	Intervention Group (*n* = 14)			Control Group (*n* = 13)		
	Baseline	After 8 Weeks	*p*-Value *	Baseline	After 8 Weeks	*p*-Value *
Anthropometric data						
Body weight, Kg	86 (80.5–90.5)	85.3 (77–88.3)	0.003	82 (70–87.5)	81 (70–86.5)	0.233
BMI, Kg/m^2^	31.3 (27.5–35)	30.2 (27.3–34.7)	0.003	27.1 (25.1–35.9)	27.1 (25.1–35.6)	0.237
Neck circumference, cm	40 (37–41.5)	38.8 (37.4–40.5)	0.056	38 (37–41.5)	40 (36.3–41.3)	0.878
Waist circumference, cm	108.5 (102–114)	107 (99–113.5)	0.022	107 (97–114.5)	106 (99.5–115)	0.791
Hip circumference, cm	97 (93.5–109)	98.5 (92.8–108.8)	0.514	96 (91.5–107.5)	95 (92.3–106)	0.150
Polysomnographic data						
AHI/h	30.5 (22.5–41.3)	34.5 (14.5–45)	0.875	37 (25.5–43.5)	43 (28–46)	0.529
CT90	12 (4–21.3)	7 (3.8–23.3)	0.423	22 (3.5–28.5)	13 (5.6–25)	0.625
ODI/h	30 (22.3–39.8)	30.5 (14.8–45.3)	0.780	34 (29.5–44.5)	38 (25.5–44)	0.944
Exercise Capacity						
6MWD, m	548.2 ± 83.9	567.1 ± 85.3	0.020	529.8 ± 102.7	519.8 ± 108.0	0.249
Questionnaires						
ESS	8 (3–13)	8 (4–10.3)	0.634	8 (6.5–11)	7 (6.5–10.5)	0.500
QSQ						
Total score	180.5 (151–200)	186 (153.8–201)	0.572	172 (133–186.5)	176 (130–200)	0.388
Daytime Sleepiness	38 (29.3–40.3)	37 (34.3–40)	0.422	36 (29.5–37.5)	36 (29.5–39.5)	0.916
Diurnal symptoms	59.5 (45–66)	56 (48.5–61.3)	0.889	50 (43.5–62)	53 (36.5–63)	0.937
Nocturnal symptoms	35 (29.8–42.3)	39 (32.3–42.5)	0.170	35 (26.5–38)	35 (30–41.5)	0.116
Emotions	27 (21.8–29.3)	28.5 (24.5–30.5)	0.169	30 (19.5–31)	30 (19.5–32)	0.641
Social interactions	23 (21–25)	21.5 (18.8–24.3)	0.431	23 (17–24)	23 (19.5–25.5)	0.372
HAD						
Total score	9 (3.8–18.8)	10 (5.8–15.6)	0.783	14 (6–23.5)	12 (6–24)	0.350
Anxiety	6 (2.8–10)	7.5 (4–9.3)	0.916	8 (6–12.5)	7 (4.5–12)	0.126
Depression	2.5 (1–8)	3 (1.8–6.5)	0.587	6 (1–9)	6 (1.5–11.5)	0.142

Abbreviations. BMI: body mass index, AHI: Apnea Hypopnea Index, CT90: cumulative percentages of time spent at saturations below 90%, ODI: oxygen desaturation index, 6MWD: Six-minute walking distance, ESS: Epworth Sleepiness Scale; QSQ: Quebec Sleep Questionnaire, and HAD: Hospital Anxiety and Depression Score. * Intragroup comparison before and after using paired *t*-test for continuous variables and the Mann–Whitney *U* test if the distribution was skewed.

**Table 3 jcm-08-00361-t003:** Effects of combined intervention, analyzed by age (i.e., younger or older than 60).

	Group < 60 years (*n* = 6)			Group > 60 years (*n* = 8)		
	Baseline	After 8 Weeks	*p*-Value *	Baseline	After 8 Weeks	*p*-Value *
Anthropometric data						
Body weight, kg	90 (81.3–101.5)	86.5 (77–100.5)	0.042	84 (71.5–86.8)	82.6 (71–86.6)	0.027
BMI, kg/m^2^	33.1 (28.7–37)	32.1 (27.6–35.9)	0.043	29.6 (25–34)	29.3 (25–33.4)	0.028
Neck circumference, cm	38.5 (35.8–40.3)	38.8 (35.8–40)	0.655	40.5 (37.8–44.5)	38.5 (37.6–44.3)	0.072
Waist circumference, cm	113 (102.3–125.8)	113.5 (98.5–123.3)	0.144	108 (100.3–109.8)	106.5 (100.3–107.8)	0.088
Hip circumference, cm	97 (88.8–111)	92 (87.5–112.5)	0.414	99.5 (95.5–113)	101 (94.8–110.3)	0.833
Polysomnographic data						
AHI/h	29.5 (21.8–48.3)	15.5 (11–34)	0.028	31.5 (21–38.8)	43.5 (24.8–48)	0.017
CT90	4.5 (3.8–10)	4 (2.5–11)	0.498	14.5 (11.5–21.8)	16.5 (6–51)	0.173
ODI/h	25 (15–35)	15.5 (12.5–32.5)	0.223	33 (25.8–43.3)	41.5 (24.3–46.8)	0.441
Exercise capacity						
6MWD, m	585.0 ± 63.0	597.8 ± 70.1	0.173	520.7 ± 90.6	544.0 ± 92.7	0.036
Questionnaires						
ESS	9 (6.8–19)	10.5 (4–19.5)	0.666	6.5 (3–11.8)	7.5 (2.3–8.8)	0.523
QSQ						
Total score	185.5 (150.8–201)	171.5 (142.5–194.5)	0.046	177 (136.3–200)	186.5 (170–201)	0.069
Daytime Sleepiness	38.5 (28–40.3)	37.5 (25.8–40)	0.074	35 (25.8–40.3)	37 (35–41)	0.058
Diurnal symptoms	61 (45–66)	52 (45.8–62.5)	0.223	56.5 (39.3–65.5)	58.5 (49–61.8)	0.233
Nocturnal symptoms	38 (32.3–44)	38 (32.3–44.3)	0.914	35 (25.3–38.3)	40 (32–41.8)	0.061
Emotions	26 (21.5–30)	26.5 (20–30.3)	0.683	27.5 (21.3–29.8)	30 (26.3–31.5)	0.040
Social interactions	23 (19.8–25.5)	19 (13.8–23.5)	0.116	23 (19–24.8)	22.5 (20.3–24.8)	0.526
HAD						
Total score	9.5 (2.8–25)	11 (7.8–24.3)	0.236	9 (5–16)	8 (5.3–15)	1.000
Anxiety	8 (2–11.8)	8 (6–11.8)	0.750	5.5 (3.5–9.25)	5 (4–9)	0.932
Depression	1.5 (0.8–11.8)	3 (1.8–11)	0.285	3.5 (1.3–7)	3 (1.3–6)	0.829

Abbreviations. BMI: body mass index, AHI: Apnea Hypopnea Index, CT90: cumulative percentages of time spent at saturations below 90%, ODI: oxygen desaturation index, 6MWD: Six minute walking distance, ESS: Epworth Sleepiness Scale; QSQ: Quebec Sleep Questionnaire, and HAD: Hospital Anxiety and Depression Score. * Intragroup comparison before and after intervention using a paired *t*-test for continuous variables and a Mann–Whitney *U* test for skewed distributions.

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
