# Peer review of "Effects of a Combined Community Exercise Program in Obstructive Sleep Apnea Syndrome: A Randomized Clinical Trial"

_jcm, 2019, doi:10.3390/jcm8030361_

Round 1

Reviewer 1 Report

The authors describe a randomized controlled study in men with moderate to severe OSA with one group receiving an urban walking exercise program, oropharyngeal exercises, diet and sleep recommendations and the other diet and sleep recommendations for 8 weeks. The small numbers chosen are based on AHI reduction of 8 based on a study that chose physical activity as the only intervention.

Major

The use of multiple interventions to assess an AHI effect (based on a study that chose only physical activity - Kline et al) makes it difficult to assess which of the interventions caused the AHI reduction. This is especially true given the extremely small number of subjects randomized to each arm of the study. If the primary intent was to show an effect on AHI reduction independent of weight change, the study should have been appropriately designed so with larger numbers so that AHI decrease can be assessed in those with and without weight loss. This study was not adequately powered to assess the benefit of a multipronged approach beyond that of the benefit accrued from weight loss which is what happened in the <60 age group

There is quite a bit of discussion about how the weight change in less than 60 group leads to reduction in AHI whereas in those above 60 did not have a reduction in AHI despite the weight loss. While this is an important finding, this was not the primary goal of the study and the study should have been appropriately designed to research this hypothesis. The <60 group had a higher BMI and it not clear based on the results as to what the degree of compliance with the walking program was (how many subjects achieved target heart rate in their walks, how many were adherent to their walking schedules), what kind of dietary recommendations were made and how many were followed, how many hours of sleep occurred in each group (an actigraphy would have been useful) etc.

The authors fail to adequately explain why there was a lack of change in AHI in >60 years group despite the weight reduction. Were they less adherent to the exercise program (both oropharyngeal and urban exercise program), was it because of night-to-night variability or more supine sleep in follow up sleep study etc. in the elderly group?

While a detailed explanation of the exercise program is given, the nature of dietary and sleep recommendations are not mentioned.

Minor

They should state clearly that none of the subjects were on CPAP during this study.

Reasons why only men were chosen is not clear. Whether they were all non-smokers is also not clear. Medications and comorbidities are not detailed. A number of these confounders may also affect study results especially in those above 60

Discussion about muscle innervation and the sensitivity of ventilatory system in younger subjects while interesting should be limited, given that this study was not aimed to assess changes in these aspects from either weight loss or other exercises. 

Author Response

Response to Reviewer 1 Comments

Point 1: Major. The use of multiple interventions to assess an AHI effect (based on a study that chose only physical activity - Kline et al) makes it difficult to assess which of the interventions caused the AHI reduction. This is especially true given the extremely small number of subjects randomized to each arm of the study. If the primary intent was to show an effect on AHI reduction independent of weight change, the study should have been appropriately designed so with larger numbers so that AHI decrease can be assessed in those with and without weight loss. This study was not adequately powered to assess the benefit of a multipronged approach beyond that of the benefit accrued from weight loss which is what happened in the <60 age group

Response 1: Thanks for this comment. In 2014, when we presented the project to the SEPAR-ALAT grant, we calculated the sample size with three similar protocols: Kline (11 each arm) and Norman (7 each arm) for physical training; and Guimaraes (10 each arm) for oropharyngeal exercises. We only add in the methods the value most exigent. We assume our error, and we added the value for Guimaraes et al. in the “experimental section/ statistical analysis”. 

The reviewer is right when he says that our sample size is small. This fact is due that the analysis of the difference in age range, was not our primary objective. We expected the same behavior that the other authors, but surprisingly, we have many patients older than 60 years with different results of the other authors. Moreover, we compared with the previous articles and all had a population under 60 years (Kline, Sengul, Servantes, Barnes, Norman, Guimaraes, Ieto, and Tang. In the case of Desplan et al., they do not describe the mean age, only the range 35-70 y). At that moment we decided to report this fact, because “open a new unexplored door”: the effects of exercise in patients older than 60 years.

Aware of this limitation, we added the sentence in the limitation section “On the other hand, the sample size was small. We design a protocol with sample size calculation to explore the results with a comprehensive program in the general population without age categories, and we do not expect found differences between patients upper or below 60 years. We think that in the following protocols it is necessary to investigate specifically in older adults.

Point 2: Major. There is quite a bit of discussion about how the weight change in less than 60 group leads to reduction in AHI whereas in those above 60 did not have a reduction in AHI despite the weight loss. While this is an important finding, this was not the primary goal of the study and the study should have been appropriately designed to research this hypothesis. The <60 group had a higher BMI and it not clear based on the results as to what the degree of compliance with the walking program was (how many subjects achieved target heart rate in their walks, how many were adherent to their walking schedules), what kind of dietary recommendations were made and how many were followed, how many hours of sleep occurred in each group (an actigraphy would have been useful) etc.

Response 2: Thanks for this comment. For us was a complex aspect, because we supervised 50% of the sessions, and we called by phone to ask for the other sessions. When one patient could not go to a session (supervised or unsupervised), we reprogrammed to another schedule the visit to the same week or the following week. All patients reported compliance of 100%. We added a paragraph in “experimental section/intervention” explain this point. However, regarding the heart rate target, we controlled in the all supervised session, but not in the unsupervised sessions. To get patients to learn to reach the required level, we performed most of the supervised sessions at the beginning of the protocol. When the patient performed an unsupervised session, we insisted that they reach the required level.

Concerning the dietary and sleep recommendations, we added more information in order to clarify this point in the experimental section and added a document used in Appendix 1.

Finally, although there was a difference in the initial weight between subjects over and under 60, this difference was not significant. In any case, both groups significantly decreased their weight and BMI post-intervention. In this area knowledge is limited, so we do not venture to raise a cause. What we do propose and believe is important is that the apneas of both groups are different from the functional point of view, since they have similar AHI, but patients over 60 years of age desaturate three times more than patients under 60 years.

We added a paragraph in the section “discussion” about this topic.

Point 3: Major. The authors fail to adequately explain why there was a lack of change in AHI in >60 years group despite the weight reduction. Were they less adherent to the exercise program (both oropharyngeal and urban exercise program), was it because of night-to-night variability or more supine sleep in follow up sleep study etc. in the elderly group?

Response 3: Thanks for this comment. We added a paragraph in the section discussion explain the change of BMI in both groups with a reduction of AHI only in the <60. We do not think that the problem was the adherence, because in general exercise when one patient could not go to a session (supervised or unsupervised), we reprogrammed to another schedule the visit to the same week or the following week. In the oropharyngeal session, the patients completed a compliance chart every day. If one day the patient does not do the exercises, he could recover one day of the weekend. All patients reported 100% of compliance.

It is correct that the night-to-night variability and the supine position can modify the results in one sleep study. However, all patients had the same possibility. We made only one exam based on the majority of previous reports. However, we analyzed during the planning phase the feasibility of taking more exams but was not possible. In any case, we agree with the reviewer, and we think it is important to add it to the limitations of the study.

Point 4: Major. While a detailed explanation of the exercise program is given, the nature of dietary and sleep recommendations are not mentioned.

Response 4: Thanks for this comment. We added a paragraph in the methods section with the description of general recommendation; additionally, we added the document used like supplementary information.

Point 5: Minor. They should state clearly that none of the subjects were on CPAP during this study.

Response 5: We agree that it is not clear. We added a sentence in experimental section/ participants with the information.

Point 6: Minor. Reasons why only men were chosen is not clear. Whether they were all non-smokers is also not clear. Medications and comorbidities are not detailed. A number of these confounders may also affect study results especially in those above 60

Response 6:  Thanks for the suggestion. Not all patients were men, only 15, this information is included in results and table 1 (divided by groups).

We excluded all smokers and patients with derivate diseases (COPD for example) due to the role of tobacco in the muscle system. Unfortunately, We forgot to put it in the exclusion criteria. We added in methods.

When we planned this project, we analyzed the effect of comorbidities and the medications in the cardiovascular, respiratory and muscular response to the exercise deeply. We do not include the comorbidities or medicaments in the first moment because we had strict inclusion criteria and we analyzed all the medications of patients specifically looking for drugs that have a mechanism of action at the muscular level. Later in the analysis, we do not found differences between groups.

Now, We added a paragraph in the section results including the comorbidities in “number of comorbidities” and the comorbidities founded.

Point 7: Minor. Discussion about muscle innervation and the sensitivity of ventilatory system in younger subjects while interesting should be limited, given that this study was not aimed to assess changes in these aspects from either weight loss or other exercises.

Response 7: We agree. We eliminated some sentences from the paragraph of Saboisky that were redundant. We only left the big ideas. Also, we eliminated one sentences of the paragraph of Edward.

Best regards

Rodrigo Torres-Castro

Reviewer 2 Report

General considerations:

Though on the whole of an acceptable standard, some language improvement is required. I have provided some examples from the 1st two pages:

Line 28/29: Better: `However, in intervention patients younger than 60 (n = 6) a reduction of the apnea-hypopnea index from 29.5 [21.8-48.3] to 15.5 [11-34] events/h (p = 0.028) was observed.´

Line 30: Better: . . . modify the apnea-hypopnea index . . .

Line 38: Better: In obstructive sleep apnea (OSA) intermittent collapse of the upper airway occurs during sleep, . . .

Or: Obstructive sleep apnea (OSA) is characterized by intermittent collapse . . .

Line 39: Better: This disease affects 9% to 38% of the general . . .

Or: This disease affects between 9% and 38% of the general . . .

Line 44: Better: . . . morbidity and mortality from cardiovascular . . .

Or: due to

Or: resulting from

Specific comments:

Line 19-33: I would suggest mentioning the number of patients included in the abstract.

Line 46: Though reference 9 does include results on AHI reduction, the actual reference (with respect to AHI reduction) should be the article by the same group on the same study population from 2011: `The Effect of Exercise Training on Obstructive Sleep Apnea and Sleep Quality: A Randomized Controlled Trial´.

Line 48/49: I am not sure that references 15/16 demonstrate reduced OSA prevalence with exercise. However, this may be found in the 2004 article by Peppard and Young from the Wisconsin cohort: `Exercise and Sleep-Disordered Breathing: an Association Independent of Body Habitus.´

Line 55: It would perhaps also be worthwhile to mention the recent 2018 meta-analysis of Mendelson et al. in this context: Obstructive Sleep Apnea Syndrome, Objectively Measured Physical Activity and Exercise Training Interventions: a Systematic Review and Meta-analysis.

Line 57-60: I would suggest to continue the elaboration on causes of OSA improvement in spite of unchanged BMI as continued in line 61-65 and move lines 57-60 to after line 65. In this context of causes of OSA improvement, the above mentioned article by Mendelson has provided a very nice figure.

Line 105: Unless the recommendations are identical, it should be reference 26 (instead of 24).

Line 117 : Reference 25 does not refer to the AASM scoring guidelines. Besides, if frontal if frontal electrodes were not used, the PSG is not entirely compliant with AASM guidelines. Though this is not really important, it should be mentioned.

Line 112-135: I am not quite clear as to why a PSG as well as a PG were performed. Was the study before intervention a PSG and after intervention a PG? If so, this should be mentioned more clearly, as this could have implications for the results and should also be discussed in the limitations section.

Line 180 and 191: Tables 1 and 2: There is some discrepancy between these tables, particularly regarding baseline values for body weight for control as well intervention group. Kindly check these (and the other values too). Hopefully, the body weight values in table 2 are the correct ones, otherwise there might be some problem with weight loss results.

Line 229: I suppose it should be table 3 (and not table 2).

Line 242: Reference 30 refers to the Spanish validation of the ESS. Kindly provide the correct reference to prove the point in question.

Line 243-253: The findings of Saboisky et al. and Andersen may explain why OSA in the elderly may be a different phenotype due to loss of motor units and muscle fibers, which is partially compensated (during wakefulness at least) by a higher amplitude of motor unit potentials. But, why does this serve to explain why the upper airway muscles in the elderly may not be trainable? Sarcopenia in the elderly is found in other (peripheral) muscles too, but in these muscle strength and function can definitely be trained in the elderly too. In my mind, the highly interesting results in this article  regarding the elderly are not yet explained.

Line 275-284: I think the small sample sizes should be mentioned as limitation.

Though it is quite clear that the current article attempts a real life approach, I would discuss somewhere that it is completely unclear to which specific intervention the noted effects are attributable.

Author Response

Response to Reviewer 2 Comments

Point 1: Though on the whole of an acceptable standard, some language improvement is required. I have provided some examples from the 1st two pages:

Line 28/29: Better: `However, in intervention patients younger than 60 (n = 6) a reduction of the apnea-hypopnea index from 29.5 [21.8-48.3] to 15.5 [11-34] events/h (p = 0.028) was observed.´

Response 1: Thanks for this comment. We changed the phrase by the suggestion of reviewer 2.

Point 2: Line 30: Better: . . . modify the apnea-hypopnea index . . .

Response 2: We agree. We added the word by the suggestion of reviewer 2.

Point 3: Line 38: Better: In obstructive sleep apnea (OSA) intermittent collapse of the upper airway occurs during sleep, . . .

Or: Obstructive sleep apnea (OSA) is characterized by intermittent collapse .

Response 3: Thanks for this suggestion. We changed the phrase by the 1st suggestion of reviewer 2.

Point 4: Line 39: Better: This disease affects 9% to 38% of the general . . .

Or: This disease affects between 9% and 38% of the general . . .

Response 4: We changed the phrase by the 2nd suggestion of reviewer 2.

Point 5: Line 44: Better: . . . morbidity and mortality from cardiovascular . . .

Or: due to

Or: resulting from

Response 5: We changed the phrase by the 2nd suggestion of reviewer 2.

Point 6: Line 19-33: I would suggest mentioning the number of patients included in the abstract.

Response 6: We included the total number of patients enrolled, randomized and finally included. Thanks for this comment.

Point 7: Line 46: Though reference 9 does include results on AHI reduction, the actual reference (with respect to AHI reduction) should be the article by the same group on the same study population from 2011: `The Effect of Exercise Training on Obstructive Sleep Apnea and Sleep Quality: A Randomized Controlled Trial´.

Response 7: Thanks for this valuable comment. We made a mistake with the reference manager (endnote). We replaced the reference by the correct article.

Point 8: Line 48/49: I am not sure that references 15/16 demonstrate reduced OSA prevalence with exercise. However, this may be found in the 2004 article by Peppard and Young from the Wisconsin cohort: `Exercise and Sleep-Disordered Breathing: an Association Independent of Body Habitus.´

Response 8: Thanks for this comment. As written, We agree with you. We think that our idea is not clear, and we replaced the sentence “regular physical activity” with “physical training programmes.” The references of Iftikhar (15) is a systematic review and metanalysis and describe the reduction of the AHI and the sleepiness with physical training programmes, not with regular physical activity. We think that is an influential article because include five different articles that showed the diminution of these outcomes. Additionally, We included a new reference, suggest in the following point (Mendelson).

Point 9: Line 55: It would perhaps also be worthwhile to mention the recent 2018 meta-analysis of Mendelson et al. in this context: Obstructive Sleep Apnea Syndrome, Objectively Measured Physical Activity and Exercise Training Interventions: a Systematic Review and Meta-analysis.

Response 9: We are very thankful for this comment (really!), because We did not know this article and after reading it, We see that it provides additional information to improve other comments and update the manuscript.

Point 10: Line 57-60: I would suggest to continue the elaboration on causes of OSA improvement in spite of unchanged BMI as continued in line 61-65 and move lines 57-60 to after line 65. In this context of causes of OSA improvement, the above mentioned article by Mendelson has provided a very nice figure.

Response 10: We agree with you. We moved the lines in the order suggested. We corrected the numbers of references.

Point 11: Line 105: Unless the recommendations are identical, it should be reference 26 (instead of 24).

Response 11: Thanks for this comment, We reviewed the order of references and effectively is reference 26. During the writing process, we modified a reference and modified the number of the last ones. Also, we review each of the following references one by one

Point 12: Line 117 : Reference 25 does not refer to the AASM scoring guidelines. Besides, if frontal if frontal electrodes were not used, the PSG is not entirely compliant with AASM guidelines. Though this is not really important, it should be mentioned.

Response 12: Thanks for this comment. We reviewed the references and replaced 25 by 27 reference. Additionally, We added the sentences “frontal electrodes were not used”.

Point 13: Line 112-135: I am not quite clear as to why a PSG as well as a PG were performed. Was the study before intervention a PSG and after intervention a PG? If so, this should be mentioned more clearly, as this could have implications for the results and should also be discussed in the limitations section.

Response 13: In order to clarify this point, we added the following sentence: “All patients were evaluated through the same test (Polysomnography or Respiratory polygraphy) before and after the intervention. The choice of which test was chosen for each patient was made by the pulmonologist's medical criteria.”. The additional variables delivered by the PSG were not analyzed.

Point 14: Line 180 and 191: Tables 1 and 2: There is some discrepancy between these tables, particularly regarding baseline values for body weight for control as well intervention group. Kindly check these (and the other values too). Hopefully, the body weight values in table 2 are the correct ones, otherwise there might be some problem with weight loss results

Response 14: Thank you to the reviewer. Probably, the mistake was due the change of order of the columns of the table. We corroborated the results in the SPSS and the table 2 had the correct results. In the table 1 We corrected the values.

Point 15: Line 229: I suppose it should be table 3 (and not table 2).

Response 15: Thank you for this comment. I replace the number 2 by the number 3

Point 16: Line 242: Reference 30 refers to the Spanish validation of the ESS. Kindly provide the correct reference to prove the point in question.

Response 16: Thanks for this correction. We reviewed, and effectively this text is based in results of Edward (reference 34)

Point 17: Line 243-253: The findings of Saboisky et al. and Andersen may explain why OSA in the elderly may be a different phenotype due to loss of motor units and muscle fibers, which is partially compensated (during wakefulness at least) by a higher amplitude of motor unit potentials. But, why does this serve to explain why the upper airway muscles in the elderly may not be trainable? Sarcopenia in the elderly is found in other (peripheral) muscles too, but in these muscle strength and function can definitely be trained in the elderly too. In my mind, the highly interesting results in this article  regarding the elderly are not yet explained.

Response 17: Thanks for this comment. We agree with you.

When we re-read the original sentence, appear to be very categorical this sentence. We conserved the objective values founded by Saboisky et al. However We deleted our interpretation because we do not considered the general musculature. We added a paragraph with information about our results in the physical capacity in the group older than 60 and we compared with the literature of physical training programs in sarcopenia.

Additionally, we emphasize the lack of evidence in people older than 60, because all papers explore in people younger than 60 in both types of exercise (general training and oropharyngeal exercises).

Point 18: Line 275-284: I think the small sample sizes should be mentioned as limitation.

Response 18: Thanks for this comment. We added a sentence in the limitation. When we calculated the sample size, we analyzed a comprehensive program in adults, independent of age. Surprisingly, we found differences between age range.

Point 19: Though it is quite clear that the current article attempts a real life approach, I would discuss somewhere that it is completely unclear to which specific intervention the noted effects are attributable.

Response 19: Thanks for this comment. When we planned this protocol, we knew that the oropharyngeal exercises and general exercises were effective in reducing the AHI. At that moment we designed a combined program because it had not been reported. However, We knew that an inpatient program was difficult to carry out massively due to the high prevalence of this disease. The Hospital Clínic of Barcelona cares for a large number of patients, and we had to look for an alternative for those patients who had moderate OSA and who had poor adherence to CPAP and had recently been validated circuits in the parks that have been used with great success in patients with COPD. We did not expect to recruit so many older patients. However, we decided to report it because we realized that all the other articles used young groups of patients with similar results to our patients under 60, and especially because it opens an “unexplored door”: the effect of these exercises in a population over 60.

We think that we need to return to study the effects of individual interventions, but now in a population over 60 years.

We added in limitation the necessity to study the interventions alone in patients over 60y.

Best regards

Rodrigo Torres-Castro
